# Hypertriglyceridemia Is Associated with More Severe Histological Glomerulosclerosis in IgA Nephropathy

**DOI:** 10.3390/jcm10184236

**Published:** 2021-09-18

**Authors:** Won Jung Choi, Yu Ah Hong, Ji Won Min, Eun Sil Koh, Hyung Duk Kim, Tae Hyun Ban, Young Soo Kim, Yong Kyun Kim, Seok Joon Shin, Seok Young Kim, Chul Woo Yang, Yoon-Kyung Chang

**Affiliations:** 1Department of Internal Medicine, Daejeon St. Mary’s Hospital, College of Medicine, The Catholic University of Korea, Daejeon 34943, Korea; 1jungchoi@gmail.com (W.J.C.); amorfati@catholic.ac.kr (Y.A.H.); alterego54@catholic.ac.kr (S.Y.K.); 2Department of Internal Medicine, Bucheon St. Mary’s Hospital, College of Medicine, The Catholic University of Korea, Bucheon 14647, Korea; blueberi12@gmail.com; 3Department of Internal Medicine, Yeouido St. Mary’s Hospital, College of Medicine, The Catholic University of Korea, Seoul 07345, Korea; fiji79@catholic.ac.kr; 4Department of Internal Medicine, Seoul St. Mary’s Hospital, College of Medicine, The Catholic University of Korea, Seoul 06591, Korea; scamph@catholic.ac.kr (H.D.K.); yangch@catholic.ac.kr (C.W.Y.); 5Department of Internal Medicine, Eunpyeong St. Mary’s Hospital, College of Medicine, The Catholic University of Korea, Seoul 03476, Korea; deux0123@catholic.ac.kr; 6Department of Internal Medicine, Uijeongbu St. Mary’s Hospital, College of Medicine, The Catholic University of Korea, Uijeongbu 11765, Korea; dr52916@catholic.ac.kr; 7Department of Internal Medicine, St. Vincent’s Hospital, College of Medicine, The Catholic University of Korea, Suwon 16247, Korea; drkimyk@catholic.ac.kr; 8Department of Internal Medicine, Incheon St. Mary’s Hospital, College of Medicine, The Catholic University of Korea, Incheon 22711, Korea; imkidney@catholic.ac.kr

**Keywords:** hypertriglyceridemia, dyslipidemia, IgAN

## Abstract

IgA nephropathy (IgAN) is a globally well-known primary glomerular nephropathy. Hypertriglyceridemia (HTG) is one factor contributing to atherosclerosis and is a common complication of renal failure. HTG is a significant risk factor for decreased renal function in patients with IgAN. We evaluated the association of HTG with the histopathological features of IgAN patients. A total of 480 patients diagnosed with IgAN via kidney biopsy from eight university hospitals affiliated with the College of Medicine of the Catholic University of Korea were included in the final cohort. Pathological features were evaluated by eight expert pathologists with hospital consensus. HTG was defined as a serum triglyceride (TG) level of ≥150 mg/dL. In the study population analysis, the HTG group was older, with more males; higher body mass index (BMI), low-density lipoprotein cholesterol (LDL-C) and spot urine protein ratio; and lower estimated glomerular filtration rate (eGFR). In the lipid profile analysis, eGFR was negatively correlated with TGs/ high-density lipoprotein cholesterol (HDL) and triglyceride-glucose index (TyG). Proteinuria positively correlated with TGs/HDL, non-HDL/HDL, LDL/HDL, TyG, TGs and LDL. The percentages of global sclerosis (GS), segmental sclerosis (SS) and capsular adhesion (CA), and the scores for mesangial matrix expansion (MME) and mesangial cell proliferation (MCP), were more elevated in the HTG group compared to the normal TG group. Multivariable linear regression analysis showed that the percentages of global sclerosis, segmental sclerosis and capsular adhesion, as well as the scores for mesangial matrix expansion and mesangial cell proliferation, were positively associated with TG level. In binary logistic regression, the HTG group showed a higher risk for global sclerosis and segmental sclerosis. In conclusion, HTG is a significant risk factor for glomerulosclerosis in IgAN.

## 1. Introduction

Dyslipidemia, defined as high levels of triglycerides (TGs), low-density lipoprotein cholesterol (LDL-C) and total cholesterol (TC), or a low level of high-density lipoprotein cholesterol (HDL-C), is a common complication of renal failure, which is associated with mortality in patients with CKD [1,2]. Dyslipidemia and hypertriglyceridemia (HTG) can exacerbate the progression of kidney dysfunction [3]. The ratios of lipid profiles, especially the TG/HDL ratio and triglyceride-glucose (TyG) index, are measures of insulin resistance [4]. The TG/HDL ratio is related to the progression of proteinuria and the prediction of CKD. Additionally, the TyG index is a predictor of CKD incidence [5,6]. Triglyceride-rich and apolipoprotein B(apoB)-containing lipoproteins are associated with decreased renal function [7].

Dyslipidemia causes glomerulosclerosis and promotes endothelial and mesangial proliferation [8,9,10]. The mechanism of the effect of dyslipidemia or HTG on glomerular change is not understood. However, the accumulation of lipids in podocytes plays a major role in HTG-associated renal injury [11].

IgA nephropathy is the most common glomerulonephritis disease worldwide. In Korea, it accounts for more than half of all cases of primary glomerulonephritis [12]. 

HTG is a significant risk factor for decreased renal function in patients with IgA nephropathy [13]. Additionally, dyslipidemia and serum triglyceride level are significantly associated with more prominent renal progression in chronic glomerulonephritis [14].

Various clinical factors affect IgAN prognosis. However, few studies consider the histopathological features of IgAN with HTG and related lipid profiles. 

In this study, we compared the histopathological features of IgAN in an HTG group and a normal TG group. 

## 2. Materials and Methods

### 2.1. Study Design and Data Source

This study is a cross-sectional review of a multi-center cohort of patients who underwent kidney biopsies at eight university hospitals affiliated with the College of Medicine of the Catholic University of Korea between January 2015 and May 2020 and who were diagnosed with IgAN. We excluded patients aged <18 years, those diagnosed with diabetes mellitus, those with no available serum triglyceride and those whose histologic information did not adhere to the standard description form or indicate sufficient glomeruli for diagnosis. A total of 480 patients were enrolled in the final cohort. The patients were divided into two groups according to the following diagnostic criteria: group 1—TG <150 mg/dL; group 2—TG ≥150 mg/dL [4,5,6]. Figure 1 shows the flow chart for this study population.

### 2.2. Data Collection Definitions and Measurement

Using the database of the Kidney Biopsy Registry of the Catholic Medical Center, data collected at the time of the kidney biopsies were reviewed. Baseline clinical data and laboratory data were collected. Proteinuria was assessed via the spot urine protein to creatinine ratio. The estimated glomerular filtration rate (eGFR) was calculated with the equation from the Modification of Diet in Renal Disease study [15]. The degree of urinary red blood cell (RBC) sedimentation was divided into five categories, as follows: <3 RBCs/high-power field (HPF), 0; 3–5 RBCs/HPF, 1; 5–9 RBCs/HPF, 2; 10–19 RBCs/HPF, 3; more than 19 RBCs/HPF, 4. 

The lipid-related ratio was defined based on the TGs, LDL-C, HDL-C and total cholesterol, measured at the serum level. Non-HDL-C was calculated as the level of HDL-C subtracted from the TC. The three lipid-related ratios of TGs (mg/dL), LDL-C (mg/dL) and non-HDL-C (mg/dL) were divided by HDL-C (mg/dL). The TyG index was calculated as ln (fasting triglycerides (mg/dL) × fasting plasma glucose (mg/dL)/2) [6]. We also assessed treatment plans after the diagnosis of IgAN, including use of anti-hypertensive medications, lipid-lowering agents and immunosuppressive medications, including steroids.

### 2.3. Histopathologic Parameters

Renal pathology was reviewed by expert renal pathologists in each of the eight centers. Using light microscopy, global and segmental glomeruloscleroses, capsular adhesion and crescents were assessed along with the severity of other kidney structures: mesangial matrix expansion (MME), mesangial cell proliferation (MCP), endocapillary proliferation (ECP), interstitial fibrosis (IF), tubular atrophy (TA), arterial intimal hyalinosis, monocyte infiltration and neutrophil infiltration were graded from 0 to 4 (grade 0, absent; grade 1, trace (<20%); grade 2, mild (20–40%); grade 3, moderate (40–70%); grade 4, severe (≥70%)). The renal histologic findings were scored using our center’s histologic grading system. Renal biopsy findings were assessed according to the WHO classification (class I to VI) and Oxford MEST score. Using immunofluorescence microscopy, the severity of mesangial deposition of IgA, C3 and C4d was graded as 0 (absent), +1 (trace), +2 (mild), +3 (moderate) or +4 (marked).

### 2.4. Statistical Analysis

Continuous variables between the two groups were compared using the t-test; categorical variables were compared using the Chi-square test. 

For the correlation analysis, Spearman correlation coefficients were used. To evaluate the associations of hypertriglyceridemia and histopathologic parameters with clinical and laboratory variables, linear regression analysis was performed. Logistic regression analyses were performed to estimate the odds ratios (ORs) and 95% confidence intervals (CIs) for high-grade global and segmental glomerulosclerosis, MME and MCP for the low-grade TG group. Results ≥25% were defined as high-grade glomerulosclerosis and <25% were defined as low-grade glomerulosclerosis [16]. Grades 2–4 MME and MCP were defined as high-grade, while grades 0–1 were low grade. The proportions of high- and low-grade glomerulosclerosis, MME and MCP are outlined in Appendix A. The *p*-value was considered statistically significant at <0.05. 

All statistical analyses were performed using SPSS version 23.0 software (SPSS, Inc., Armonk, NY, USA)

## 3. Results

### 3.1. Baseline Characteristics

The mean age of the total cohort of patients was 41.5 ± 14.4 years, and the mean serum TG level was 152.2 ± 124.6 mg/dL. The number of male patients was 236 (49.2%), while female patients numbered 244 (50.8%). The baseline clinical characteristics of the study were compared between the two TG groups (Table 1).

The HTG group displayed older age, male dominance and higher BMI. Blood levels of hemoglobin, fasting glucose, total cholesterol, LDL-C, serum uric acid, C3 and C4 and the spot urine protein ratio were higher in the HTG group. Systolic and diastolic blood pressure values were higher in the HTG group, while eGFR and HDL-C were lower. 

### 3.2. The Relationship between Lipid Profiles and Clinical Variables

The lipid ratio was compared between the two TG groups. The HTG group showed significantly higher TGs/HDL, non-HDL/HDL, LDL/HDL and TyG (Table 2). We evaluated the correlation of the lipid ratio with eGFR and proteinuria (spot urine P/Cr) via Pearson correlation. eGFR was significantly negatively correlated with TGs/HDL and TyG (r = −0.106 *p* = 0.022, r = −0.148 *p* = 0.001, respectively; Table 3, Appendix A). Proteinuria (spot urine P/Cr) was significantly positively correlated with TGs/HDL, non-HDL/HDL, LDL/HDL, TyG, TGs and LDL (Table 4, Appendix A)

### 3.3. Histopathologic Characteristics by TG Group

The histopathologic features of the TG groups are shown in Table 5. The percentages of global sclerosis, segmental sclerosis and capsular adhesion were more elevated in the HTG group (*p* < 0.001, *p* = 0.001, *p* = 0.004, respectively). Additionally, MME and MCP were more elevated in the HTG group, with statistical significance (*p* = 0.008, *p* = 0.007, respectively). The immunofluorescence microscopy analysis provided no relevant findings for the mesangial deposition in the two TG groups. Additionally, the distributions of the WHO classification were not significantly different between the TG groups. The distributions of the Oxford MEST classification (*n* = 57) were described in Appendix A.

### 3.4. Association of Serum TG with Histopathologic Parameters

As regards the linear regression, the univariable analysis showed that the percentages of GS, SS and CA were positively associated with the serum TG level. Additionally, the scores for MME and MCP showed positive associations with the TG level. The multivariable linear regression analysis showed that the percentages of GS, SS and CA were associated positively with the serum TG level. The MME and MCP scores were positively associated with the TG level after adjusting for the clinical and laboratory parameters of age, sex, systolic BP, BMI, Hb, uric acid, glucose, ALT, total cholesterol, HDL-C, LDL-C, eGFR, spot urine P/Cr and serum IgA level (Table 6).

Using binary logistic regression, the odds ratios of the GS, SS, MME and MCP between the two TG groups were calculated (Table 7). Group 1, with serum TG levels <150 mg/dL, was taken as a reference group for the binary logistic regression. The HTG group showed a higher risk for global sclerosis and segmental sclerosis after adjusting for age, sex and systolic BP (model 1); for model 1 plus glucose, ALT, HDL-C, total cholesterol, uric acid and spot urine P/Cr (model 2); and for model 2 plus eGFR and BMI (model 3). 

### 3.5. Treatment Patterns of TG Groups

The treatment plans were chosen within 6 months after kidney biopsy and these were compared between the two TG groups (Table 8). There were no significant differences between the two groups except steroids use. 

## 4. Discussion

In this study, we clearly showed that HTG is associated with more severe glomerular sclerosis and mesangial expansion in IgAN patients. Additionally, HTG is related to a higher risk for glomerulosclerosis in the histopathology of IgAN. 

HTG is a risk factor for atherosclerosis and the progression of renal disease [7,17]. Syrjanen et al. reported that HTG is independently associated with the progression of IgAN [13]. Wang et al. reported that HTG relates to poor renal survival in IgAN, with 50−75% glomerulosclerosis [18]. However, they did not determine whether HTG is associated with glomerulosclerosis in IgAN. 

In our study, the HTG group displayed more severe global and segmental sclerosis, capsular adhesion and mesangial expansion and proliferation. In the multivariate linear regression, the percentage of glomerular sclerosis and the grades of mesangial proliferation and capsular adhesion were positively associated with the TG level. Logistic regression demonstrated that HTG independently increased the risk for glomerular sclerosis compared with the normal TG group and considering clinical variables.

Although the molecular mechanism through which HTG and dyslipidemia aggravate glomerular sclerosis in IgAN has not been identified, dyslipidemia was shown to injure glomerular cells in various pathogeneses. Hyperlipidemia is associated with vascular injury and atherosclerosis. Additionally, lipid abnormality affects endogenous lipid metabolism, which causes glomerular damage [19]. In an animal study, hyperlipidemia and HTG influenced podocyte injury, resulting in the development of segmental sclerosis associated with secondary damage to the tubulointerstitium [11]. Additionally, oxidized LDL induces podocyte apoptosis and glomerular loss and increases proteinuria in vitro [20].

Histopathologically, focal and segmental glomerular sclerosis were shown to correlate with serum triglyceride in unilateral nephrectomy rats [10,21]. Additionally, triglyceride-rich lipoproteins caused mesangial proliferation in vitro [9]. Lipotoxicity, in which an excess of intracellular lipids stored in lipid droplets, can cause dysregulated insulin signaling, mitochondrial dysfunction, endoplasmic reticulum stress and increased renal cell apoptosis [22]. In a recent animal study, hyperlipidemia-induced free-fatty-acid dysregulation was shown to induce oxidative stress and apoptosis, which led to podocyte injury via fibrosis, inflammation and apoptosis [23]. 

The TyG index, a surrogate marker of insulin intolerance and metabolic syndrome, can predict high-risk DM nephropathy and the progression of CKD [24,25]. In our study, there was a significant difference in the TyG index between the two TG groups. In addition, the TyG index was negatively correlated with eGFR. On the other hand, the TyG index was positively correlated with proteinuria. In summary, we assume that TGs and TyG are greater influencing factors in IgAN than cholesterol and lipoproteins, which were shown to aggravate proteinuria or decrease eGFR.

Among the clinical factors, HTG is related to obesity. Obesity-related glomerulopathy causes an increased risk of mesangial matrix expansion, podocyte hypertrophy and mesangial cell proliferation [26,27]. Additionally, Wu et al. showed that high BMI was associated with a higher risk of interstitial fibrosis [28]. In our eight-center affiliation study, obesity was an independent risk factor for mesangial expansion [29]. In our study, clinical variables including BMI and HTG independently increased the risk of glomerulosclerosis.

This study had several limitations. First, it was a cross-sectional, retrospective study, so there were no long-term follow-up data. Second, the study involved eight university hospitals, and renal biopsy specimens were reviewed by eight renal pathologists; thus, observation variability was possible. However, all hospitals used identical biopsy criteria, and all pathologists were trained by the same international expert. Third, our study cannot clearly present the mechanism by which HTG and dyslipidemia induce kidney damage in IgAN. In addition, it cannot be denied that kidney damage itself has an effect on HTG and dyslipidemia. Further research is needed to understand the mechanisms by which HTG and dyslipidemia cause renal damage in IgAN.

In conclusion, HTG (TGs ≥ 150) is associated with glomerulosclerosis and mesangial proliferation. Further, HTG is related to a higher risk for global and segmental glomerulosclerosis than normal TG levels (TG < 150). In IgA nephropathy, treatments to lower TGs may be helpful in preventing the progression of glomerulosclerosis.

## Figures and Tables

**Figure 1 jcm-10-04236-f001:**
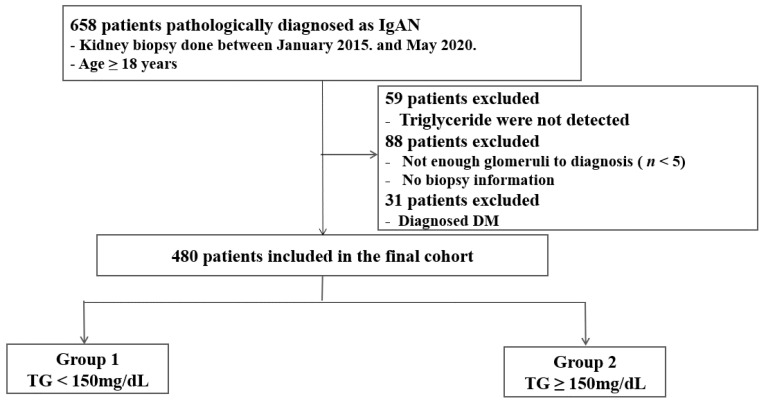
Flow diagram for study population. Abbreviation: IgAN, IgA nephropathy; DM, diabetes mellitus; TG, triglyceride.

**Table 1 jcm-10-04236-t001:** Baseline clinical variables of the two TG groups at the time of renal biopsy.

	TG			
	Total	Group 1 <150 (*n* = 313)	Group 2 ≥150 (*n* = 167)	*p **
Age (years)	41.38 ± 14.44	40.49 ± 14.99	43.41 ± 13.20	0.035
Sex (male, %)	236 (49.2)	139 (44.4)	97 (58.1)	0.004
BMI (kg/m^2^)	23.97 ± 3.71	23.04 ± 3.35	25.76 ± 3.75	<0.001
SBP (mmHg)	124.62 ± 16.51	123.30 ± 16.88	127.26 ± 15.24	0.012
DBP (mmHg)	76.29 ± 10.24	75.27 ± 10.51	78.30 ± 9.46	0.002
Hypertension (%)	158 (33.8)	100 (32.8)	58 (35.8)	0.512
Alcohol (yes, %)	88 (18.3)	53 (16.9)	35 (21.0)	0.278
Smoking (yes, %)	51 (10.6)	28 (8.9)	23 (13.8)	0.263
Triglyceride (mg/dL)	152.21 ± 124.65	92.13 ± 28.82	264.80 ± 153.99	<0.001
Total cholesterol (mg/dL)	186.45 ± 48.80	175.34 ± 38.89	206.93 ± 58.08	<0.001
LDL-C (mg/dL)	106.83 ± 39.31	104.23 ± 36.84	111.53 ± 43.40	0.057
HDL-C (mg/dL)	52.25 ± 16.85	54.86 ± 17.53	47.61 ± 14.47	<0.001
Hemoglobin (g/dL)	13.18 ± 1.94	13.00 ± 1.82	13.54 ± 2.09	0.004
hs-CRP (mg/dL)	0.60 ± 3.57	0.68 ± 4.29	0.44 ± 1.55	0.491
Glucose (mg/dL)	105.29 ± 26.12	100.62 ± 20.27	114.28 ± 32.99	<0.001
Serum creatinine (mg/dL)	1.13 ± 0.92	1.10 ± 1.00	1.18 ± 0.76	0.361
eGFR (mL/min/1.73 m^2^)	82.55 ± 33.77	85.05 ± 33.76	76.94 ± 32.07	0.011
Serum albumin (g/dL)	4.02 ± 0.58	4.05 ± 0.57	3.98 ± 0.60	0.235
AST (IU/L)	23.25 ± 11.45	22.45 ± 10.32	24.95 ± 13.30	0.023
ALT (IU/L)	22.14 ± 25.31	19.01 ± 15.37	28.30 ± 36.96	<0.001
Serum uric acid (mg/dL)	6.01 ± 1.88	5.74 ± 1.77	6.54 ± 1.99	<0.001
Urine P/Cr (mg/mg)	1.49 ± 2.48	1.08 ± 1.28	2.19 ± 3.70	<0.001
Urine RBCs (grade)	2.82 ± 1.55	2.85 ± 1.56	2.74 ± 1.55	0.485
Serum C3 (mg/dL)	105.92 ± 22.02	102.18 ± 21.77	112.76 ± 20.97	<0.001
Serum C4 (mg/dL)	28.91 ± 9.34	27.53 ± 9.00	31.28 ± 9.49	<0.001
Serum IgA (mg/dL)	312.01 ± 146.56	306.72 ± 150.57	317.49 ± 120.04	0.432

Abbreviations: BMI, body mass index; SBP, systolic blood pressure; DBP, diastolic blood pressure; LDL-C, low-density lipoprotein cholesterol; HDL-C, high-density lipoprotein cholesterol; ESR, erythrocyte sedimentation rate; hs-CRP, high-sensitivity C-reactive protein; eGFR, estimated glomerular filtration rate; AST, aspartate aminotransferase; ALT, alanine aminotransferase; RBCs, red blood cells; C3, complement 3; C4, complement 4. * *p* value is compared between TG group 1 and group 2.

**Table 2 jcm-10-04236-t002:** Lipid ratio analysis for the TG groups.

		TG		
	Total	Group 1 (<150 mg/dL) (*n* = 324)	Group 2 (≥150 mg/dL) (*n* = 187)	*p **
TGs/HDL	3.48 ± 4.12	1.89 ± 0.99	6.45 ± 5.79	<0.001
Non-HDL/HDL	2.94 ± 1.74	2.52 ± 1.49	3.73 ± 1.92	<0.001
LDL/HDL	2.15 ± 0.88	2.02 ± 0.84	2.38 ± 0.92	<0.001
TyG	4.72 ± 0.34	4.53 ± 0.20	5.09 ± 0.22	<0.001

Abbreviation: TGs, triglycerides; HDL, high-density lipoprotein cholesterol; LDL, low-density lipoprotein cholesterol; TyG, triglyceride-glucose index. * *p* value is compared between TG group 1 and group2.

**Table 3 jcm-10-04236-t003:** Pearson correlations between the lipid ratio and eGFR.

	r	*p*
TGs/HDL	−0.106	0.022
Non-HDL/HDL	−0.017	0.714
LDL/HDL	0.058	0.210
TyG	−0.148	0.001
TGs	−0.084	0.065
LDL-C	0.051	0.271
HDL-C	−0.028	0.551
Total cholesterol	0.005	0.911

Abbreviation: TGs, triglycerides; HDL, high-density lipoprotein cholesterol; LDL, low-density lipoprotein cholesterol; TyG, triglyceride-glucose index.

**Table 4 jcm-10-04236-t004:** Pearson correlations between the lipid ratio and proteinuria.

	r	*p*
TGs/HDL	0.095	0.044
Non-HDL/HDL	0.032	0.032
LDL/HDL	0.298	<0.001
TyG	0.253	<0.001
TGs	0.257	<0.001
LDL-C	0.273	<0.001
HDL-C	−0.003	0.949
Total cholesterol	0.042	0.364

Abbreviation: TGs, triglycerides; HDL, high-density lipoprotein cholesterol; LDL, low-density lipoprotein cholesterol; TyG, triglyceride-glucose index.

**Table 5 jcm-10-04236-t005:** Histopathological findings for the TG groups.

	TG		
	Group 1 <150 (*n* = 313)	Group 2 ≥150 (*n* = 167)	*p*
Light microscopy			
Global sclerosis (%)	14.95 ± 16.86	21.81 ± 23.65	<0.001
Segmental sclerosis (%)	7.56 ± 11.66	11.57 ± 14.91	0.001
Capsular adhesion (%)	7.90 ± 11.66	11.63 ± 15.01	0.004
Mesangial matrix expansion (0−4)	2.04 ± 0.85	2.26 ± 0.86	0.008
Mesangial cell proliferation (0−4)	2.03 ± 0.84	2.25 ± 0.86	0.007
Endocapillary proliferation (0−4)	0.13 ± 0.44	0.21 ± 0.58	0.074
Interstitial fibrosis (0−4)	1.34 ± 0.99	1.38 ± 0.91	0.741
Tubular atrophy (0−4)	1.31 ± 1.01	1.35 ± 0.92	0.637
Arterial intimal hyalinosis (0−4)	0.19 ± 0.60	0.27 ± 0.72	0.200
Monocyte infiltration (0−4)	1.46 ± 0.99	1.36 ± 0.89	0.314
Neutrophil infiltration (0−4)	0.07 ± 0.37	0.08 ± 0.34	0.741
Immunofluorescence microscopy			
Mesangial deposit, IgA (0−4)	3.28 ± 0.97	3.30 ± 0.97	0.794
Mesangial deposit, C3 (0−4)	2.11 ± 1.18	2.16 ± 1.15	0.660
Mesangial deposit, C4d (0−4)	0.03 ± 0.25	0.06 ± 0.34	0.348
WHO classification (*n* = 432)	*n* = 262	*n* = 141	
Class (1−6)	3.04 ± 0.84	2.96 ± 0.89	0.390

Abbreviations: TG, triglyceride; C3, complement 3; C4, complement 4; IgA, immunoglobulin A.

**Table 6 jcm-10-04236-t006:** Linear regression analysis for TGs and the histopathologic parameters.

	TG							
	Univariable				Multivariable			
	β	t	*r* ^2^	*p*	β	t	*r* ^2^	*p*
Global sclerosis	0.187	4.128	0.033	<0.001	0.173	3.544	0.060	<0.001
Segmental sclerosis	0.174	3.850	0.028	<0.001	0.149	2.995	0.107	0.003
Capsular adhesion	0.161	3.480	0.024	0.001	0.129	2.643	0.094	0.009
Mesangial matrix expansion	0.115	2.525	0.011	0.012	0.109	2.129	0.024	0.034
Mesangial cell proliferation	0.117	2.564	0.012	0.011	0.139	2.825	0.017	0.005
Endocapillary proliferation	0.039	0.845	0.001	0.398	-	-	-	-
Monocyte infiltration	0.004	0.086	0.000	0.931	-	-	-	-
Neutrophil infiltration	0.076	1.654	0.004	0.099	-	-	-	-
Interstitial fibrosis	0.054	1.182	0.001	0.238	-	-	-	-
Tubular atrophy	0.057	1.233	0.001	0.218	-	-	-	-
Arterial intimal hyalinosis	0.002	0.043	0.000	0.965	-	-	-	-
IgA mesangial deposit	0.012	0.264	0.000	0.792	-	-	-	-
C3 mesangial deposit	0.036	0.784	0.001	0.433	-	-	-	-
C4d mesangial deposit	0.000	−0.010	0.000	0.992	-	-	-	-

TG, triglyceride; AIH: arterial intimal hyalinosis; C3, complement 3; C4d, cleavage product of complement 4. Multivariable analysis was adjusted for each histologic and clinical parameter, including age, sex, systolic BP, BMI, hemoglobin, uric acid, glucose, ALT, eGFR, spot urine P/Cr, HDL-C, LDL-C, total cholesterol and serum IgA levels.

**Table 7 jcm-10-04236-t007:** Logistic regression analysis of TG groups and histopathologic parameters.

	Crude	*p*	Model 1	*p*	Model 2	*p*	Model 3	*p*
Global sclerosis								
Group 1	Ref.		Ref.		Ref.		Ref.	
Group 2	1.715 (1.131−2.601)	0.011	1.701 (1.122−2.579)	0.012	1.695 (1.055−2.723)	0.029	1.791 (1.111−2.887)	0.017
Segmental sclerosis								
Group 1	Ref.		Ref.		Ref.		Ref.	
Group 2	2.382(1.325−4.282)	<0.001	2.366 (1.316−4.253)	0.004	2.334 (1.213−4.492)	0.011	2.310 (1.200−4.446)	0.012
Mesangial matrix expansion								
Group 1	Ref.		Ref.		Ref.		Ref.	
Group 2	1.689 (0.998−2.859)	0.051	1.703 (1.006−2.882)	0.047	1.586 (0.907−2.774)	0.106	1.563 (0.893−2.737)	0.118
Mesangial cell proliferation								
Group 1	Ref.		Ref.		Ref.			
Group 2	1.587 (0.952−2.646)	0.076	1.600 (0.960−2.668)	0.072	1.409 (0.818−2.428)	0.216	1.303 (0.745−2.281)	0.353

Model 1: adjusted for age, sex and systolic BP; model 2: adjusted for model 1 + glucose, ALT, HDL-C, total cholesterol, uric acid and UPCR; model 3: adjusted for model 2 + eGFR and BMI.

**Table 8 jcm-10-04236-t008:** Treatment strategies after kidney biopsy in the TG groups.

	TG		
	Group 1 <150 (*n* = 310)	Group 2 ≥150 (*n* = 163)	*p*
Anti-hypertensive drug			
RAAS inhibitor	227 (73.2)	128 (78.5)	0.220
Calcium channel blocker	57 (18.4)	27 (16.6)	0.704
Anti-platelet agents			
Aspirin	0 (0.0)	1 (0.6)	0.346
Clopidogrel	1 (0.3)	0 (0.0)	1.000
Lipid-lowering agents			
Omega-3	60 (19.4)	28 (17.2)	0.620
Statin	92 (29.7)	52 (31.9)	0.674
Fenofibrate	3 (1.0)	2 (1.2)	1.000
Immunosuppressive agents			
Steroids	76 (24.5)	58 (35.6)	0.013
Any immunosuppressant	7 (2.20	6 (3.6)	0.389

## Data Availability

All data are reported in the article.

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
