# Peer review of "Hypertriglyceridemia Is Associated with More Severe Histological Glomerulosclerosis in IgA Nephropathy"

_jcm, 2021, doi:10.3390/jcm10184236_

Round 1

Reviewer 1 Report

This is an interesting study - I have two major comments:

  1. The histological scoring of the biopsies is out of date and really should be undertaken using the internationally accepted Oxford scoring system to allow comparability across studies.
  2. The three major confounders in this study are proteinuria, hyperglycaemia and BMI (all of which increase TGs) which were significantly different between the two groups each of which is likely to contribute to kidney disease. While I accept a logistic regression has been performed it has used not used the Oxford MEST-C score and so the results are difficult to relate to other published work.

Due to the confounders it is difficult to say which factor is driving the process:

Is glomerulosclerosis increasing proteinuria which in turn is increasing TG levels? This study does not answer this question but focusses on TGS driving glomerular scarring when in fact the raised levels may simply be a consequence of kidney injury and not a cause- this needs to be acknowledged clearly in the paper.

Author Response

1. The histological scoring of the biopsies is out of date and really should be undertaken using the internationally accepted Oxford scoring system to allow comparability across studies.

Response) We understand your comment. However, at the beginning of our cohort with eight affiliation hospitals, it was decided to use “hospital’s consensus histopathologic scoring system” for IgA nephropathy by expert renal pathologists. In this study, we used eight affiliation hospital’s consensus histopathologic scoring system, that used in previously accepted JCM article in 2020, 2021. (J. Clin. Med. 2020, 9(9), 2824; J. Clin. Med. 2021, 10(9), 1885).  In review of our cohort, the Oxford MEST classification was performed in only a few patients (n=57). Although our hospital’s consensus histopathologic scoring system is not common, compared to Oxford MEST-C classification, it can be used to find out more detail histopathological changes, because the grade system 0-4 is more stratified than other scoring systems. We analyzed the 57 patients with oxford MEST classification. The result showed there is no significant relation between MEST and TG groups. However, thesample size was too small to analyze. So, we thought that the results were unreliable. So, we did not add the following result in the manuscript. We added on Oxford MEST score in supplement table 1.

2. The three major confounders in this study are proteinuria, hyperglycaemia and BMI (all of which increase TGs) which were significantly different between the two groups each of which is likely to contribute to kidney disease. While I accept a logistic regression has been performed it has used not used the Oxford MEST-C score and so the results are difficult to relate to other published work.

Response) As we mentioned in answer 1, at the beginning of our cohort with eight affiliation hospitals, it was decided to use “hospital’s consensus histopathologic scoring system” for IgA nephropathy by expert renal pathologists, and they evaluated only a few IgAN patients using oxford MEST-C score. The number of patients was too small to be analyzed using the Oxford MEST-C score in logistic regression.

Due to the confounders it is difficult to say which factor is driving the process:

Is glomerulosclerosis increasing proteinuria which in turn is increasing TG levels? This study does not answer this question but focusses on TGS driving glomerular scarring when in fact the raised levels may simply be a consequence of kidney injury and not a cause- this needs to be acknowledged clearly in the paper.

Response) It is understandable to question presented by the reviewer. Kidney injury, glomerulopathy and molecular change can induce proteinuria and it can exacerbate dyslipidemia and elevates plasma TG levels negatively feedback (Transl Res. 2015 Apr; 165(4): 499–504). However, it is important that low serum albumin level related to hyperlipidemia and hypertriglyceridemia in proteinuria with kidney injury. (Am J Nephrol 1993;13:365–375) Although, In this study, serum albumin level was lower in TG group2, there was no statistically significant difference. In this reason, we thought that proteinuria and kidney damage could not be the major factors for HTG or dyslipidemia. However, our assumption cannot completely rule out the reviewer’s questionnaire. So, we added the limitation for this study about this perspective in discuss section.

Reviewer 2 Report

Comment to the authors:

This is an interesting study worth to report. The following are my comments.

  1. Please add “ in IgA nephropathy” in the final sentence of the abstract.
  2. Too many abbreviations in the abstract. At least, TyG, MME and MCP are not those commonly used.
  3. Patients with diabetes without any histopathological evidence of diabetic nephropathy should not be excluded from study population. They are important part of patient group of IgA nephropathy. Please provide % of diabetes status in Table 1.
  4. Please provide information for therapeutic drugs already used for dyslipidemia and IgA nephropathy, including RAS inhibitors or antiplatelets, at the time of biopsy diagnosis.
  5. The order of variables in Table 1 should be reconsidered. They should be presented by basic characteristics, including age, sex, body size, blood pressure, diabetes, hypertension, smoking status etc. first. Provide height, body weight and BMI. In the present form, BMI and blood pressure are included in “lipid profiles”.
  6. In table 4, r=0.257 for TG and proteinuria. In Figure 2C, r=0.153 for TG and proteinuria. Which is correct? Anyway, either Table or Figure is enough.
  7. Extracapillary proliferation (crescent formation) should be included in the histopathological analyses. Extracapillary proliferation is an established histopathological lesion predicting worse renal outcomes in IgA nephropathy.

Author Response

1. Please add “in IgA nephropathy” in the final sentence of the abstract.

Response) As reviewer’s mentioned, added “in IgA nephropathy” in abstract.

2. Too many abbreviations in the abstract. At least, TyG, MME and MCP are not those commonly used.

Response) In case of TyG, MME and MCP, as reviewer’s opinion, we changed the abbreviations to full terms in the abstract.

3. Patients with diabetes without any histopathological evidence of diabetic nephropathy should not be excluded from study population. They are important part of patient group of IgA nephropathy. Please provide % of diabetes status in Table 1.

Response) Diabetes strongly affect to hyperTG. Several papers showed that DM is more risk factor for dyslipidemia and hypertriglyceridemia. (J Endocr Soc. 2018 Jun 1; 2(6): 497–512; Nat Rev Nephrol . 2010 Jun;6(6):361-70.). Because diabetes can be a confounding variable of HyperTG, we excluded the patients with diabetes. So, the percentage of DM patient is not shown in this table because we excluded the DM patient.

4. Please provide information for therapeutic drugs already used for dyslipidemia and IgA nephropathy, including RAS inhibitors or antiplatelets, at the time of biopsy diagnosis.

Response) In this cohort study, we did not investigate whether the drug was used prior to the diagnosis of IgAN. However, medication information up to 6 months after the kidney biopsy were collected, and the data were added to table 8. 

5. The order of variables in Table 1 should be reconsidered. They should be presented by basic characteristics, including age, sex, body size, blood pressure, diabetes, hypertension, smoking status etc. first. Provide height, body weight and BMI. In the present form, BMI and blood pressure are included in “lipid profiles”.

Response) As reviewer commented, the order of variables was changed in the table 1.

6. In table 4, r=0.257 for TG and proteinuria. In Figure 2C, r=0.153 for TG and proteinuria. Which is correct? Anyway, either Table or Figure is enough.

Response) Table 4, the non-parametric correlation coefficient was written incorrectly, and the r=0.163 shown in Figure 2C is correct. This has been corrected. And as reviewer’s recommendation, table is remained in main text, figure is moved to the supplement.

7. Extracapillary proliferation (crescent formation) should be included in the histopathological analyses. Extracapillary proliferation is an established histopathological lesion predicting worse renal outcomes in IgA nephropathy.

Response) The pertcentage of extracapillary proliferation (crescent formation) is added to histopathological analysis. And the linear regression was performed.

Manuscript english correction for spelling and style was performed via the MDPI English editing service.

Round 2

Reviewer 1 Report

No further comments.

Reviewer 2 Report

No additional comments.